# A Spatial Optimization Model for Delineating Metropolitan Areas

**Gusiyuan Wang and Wangshu Mu ***

Faculty of Geographical Science, Beijing Normal University, Beijing 100875, China;
202321051099@mail.bnu.edu.cn
* Correspondence: muwangshu@bnu.edu.cn

**Abstract:** A metropolitan area comprises a collection of cities and counties bound by strong socioeconomic ties. Despite the pivotal role that metropolitan areas play in regional economics, their delineation remains a challenging task for researchers and urban planners. Current threshold-based delineation methods select counties based on their connection strength with prespecified core counties. Such an approach often neglects potential interactions among outlying counties and fails to identify polycentric urban structures. The delineation of a metropolitan area is fundamentally a spatial optimization problem, whose objective is to identify a set of counties with high interconnectivity while also meeting specific constraints, such as area, contiguity, and shape. In this study, we present a novel spatial optimization model designed for metropolitan area delineation. This model aims to maximize intercounty connection strength in terms of both industry and daily life. This approach ensures a more accurate representation of the multicore structure that is commonly seen in developed metropolitan areas. Additionally, our model avoids the possibility of holes in metropolitan area delineation, leading to more coherent and logical metropolitan boundaries. We provide a mixed-integer programming formulation for the proposed model. Its efficacy is demonstrated by delineating the boundaries of the Nanjing and Lhasa metropolitan areas. This study also delves into discussions and policy implications pertinent to both of these metropolitan areas.

**Keywords:** GIS; spatial optimization; metropolitan area

## 1. Introduction

A metropolitan area is a region that consists of one or more highly developed urban cores and several other closely interrelated cities and counties [1]. It first emerged in the United States in the early 20th century, when urban development led to spatial sprawl and suburbanization, creating a distinct separation between housing and employment locations. The U.S. Federal Budget Office termed this spatial organization "metropolitan districts" [2]. While these districts initially served as units for census statistics, they soon evolved into a tool for urbanization levels in a more scientific and objective manner, which aids in policy formulation and addresses challenges posed by urban expansion [3,4]. Recognizing its efficacy in addressing issues such as economic development disparities and land-use conflicts, countries such as the U.K., Japan, Canada, and China have also adopted and refined this concept [5–7].

The strong connections within metropolitan areas make it possible to promote high-quality economic growth and industry convergence between counties by breaking the constraints brought by boundaries. Consider the Greater Tokyo Area in Japan, which primarily consists of Tokyo, Kanagawa, Chiba, and Saitama, colloquially referred to as the "one capital and three counties." Tokyo stands as Japan's administrative and technological hub. Saitama, Japan's deputy capital, has absorbed certain governmental functions from Tokyo. Meanwhile, Kanagawa and Chiba serve as industrial hubs and international port centers. Each county specializes based on its inherent strengths and attributes, exerting an overall agglomeration advantage [8]. The Greater Tokyo Area spans 13,555 km², accounting for 3.5% of Japan's total land area. However, in 2018, its GDP reached USD 1.8 trillion,

representing 36% of Japan's overall GDP. Furthermore, its population stood at 36.76 million, making up 29% of Japan's total population. In addition, metropolitan areas can alleviate "urban diseases" such as housing shortages and ecological deterioration in megacities and facilitate a more convenient life for residents [9]. As a result, metropolitan areas have become new participants in the global economy due to their richer resources and stronger radiative driving effects.

Many countries have established their own metropolitan areas for planning or statistical purpose, for example, the Metropolitan Statistical Areas in the United States, the Census Metropolitan Areas in Canada, and the Functional Urban Areas in Europe. A large number of researches have been conducted in various study fields, including urban form and structure [10], climate [11], transport system [12], and sustainability development [13]. However, considering that metropolitan areas do not have established administrative boundaries, defining the scope of a metropolitan area is still the prerequisite and foundation for conducting various studies on this region.

The regionalization problem has been an important research area in GIScience and urban studies. Representative examples include political districting [14], sale territory assignment [15], and natural resource management [16]. Many related models and algorithms have been developed over more than 60 years. Most of the earlier studies applied conventional clustering and revised the result to fit spatial contiguity. With the development of technology, contiguity constraints were involved in the solution process, and heuristic models and hybrid heuristic models are developed to solve related problems [17]. However, delineating a metropolitan area is quite different. Its spatial units are usually counties or districts, which have a coarser scale than previous studies. Meanwhile, each unit has an impact on the others, making it more difficult to find the best delineation plan.

Since a metropolitan area represents a cluster of closely interconnected counties, the initial step in delineating a metropolitan area involves calculating the connection strength between these counties, a subject that has been extensively studied. The most straightforward approach for measuring the connection strength is to identify relevant socioeconomic statistical data, such as economy, population, road network, and intercounty commuting. For example, considering that commuting is the major characteristics of metropolitan areas, intercounty commuting is a way to evaluate connection strength [1]. In developing countries where actual commuting flow data is hard to acquire, the road network can be used to approximate commuting sheds [18]. Evaluation systems are established using methods such as factor analysis and principal component analysis [19–21]. This method is efficient and, consequently, it is the official delineating method for many developed countries including the U.S., Germany, and Japan [22,23].

Indirect connection strength measurement methods, such as the gravity model and the field strength model, are also applied to delineate metropolitan areas [24–27]. The core idea of this method is that the central city has a radiation effect on its surrounding region. A series of indices that can reveal the development of counties leads to a positive relationship, and distance leads to an inverse correlation. Such methods reflect a geography law that the volume of spatial interaction decreases as distance separation increases [28].

Furthermore, with the development of "space of flows" [29], more and more studies use real-time data on the flows of productive factors among cities, such as information flow, traffic flow, and financial flow [30–32]. Compared to the traditional method of using urban statistical data to depict the connections between cities, the delineation based on real-time flow data is relatively more intuitive and scientific and closer to the essence of spatial relationships.

Once the connection strength is determined, the most straightforward method to delineate a metropolitan area is by setting a threshold. This threshold can be established in various ways, such as based on the maximum number of counties included or in line with regional urban development policies. For example, a core-based statistical area (CBSA) is determined through certain thresholds of population and commuting ties in the U.S. [33]. Similarly, in approaches that identify metropolitan areas based on the amount of light

emitted at night, a light-intensity threshold is selected, and metropolitan areas can be detected by aggregating spatial units into a contiguous area [34]. However, much of the existing research primarily concentrates on the connections between central and outlying counties, given the challenge of identifying an appropriate threshold that encompasses all connections within the study area. As urbanization becomes more diffuse, cities are increasingly diversifying in their functions [35], leading to a multicore structure in metropolitan areas [36]. An effective metropolitan delineation should not only ensure tight connections between central counties and surrounding counties but also consider all potential intercounty connections.

Graph theory algorithm, such as Minimum Spanning Trees and community detection algorithms, are also widely used in urban structure research. A graph is a set of nodes and edges such that each edge connects two nodes [37]. In related studies, spatial units are represented by nodes, and the connections are denoted as edges. Metropolitan areas can be detected by subdividing the graph into subgraphs and further into local patterns with strong or loose connections between areal units, which is an effective way to identify polycentric structures in the study area [38]. However, the initial input graph is usually a subset of the complete graph, which may misjudge the connection between nodes far away and overemphasize adjacent interactions. Additionally, most of their applications are on smaller scales, like a county or single city, rather than the metropolitan area.

Another related method is head/tail breaks. Unlike conventional definitions of cities, which are imposed from the top-down subjectively by authorities and mainly based on populations, it provides a powerful tool for illustrating cities in a natural way [39]. It divides the data values into two parts around the arithmetic mean and continues the partition for values above the mean iteratively until the distribution of far more small things than large ones is violated [40]. Relevant works have been conducted using data sets such as street nodes, POIs, social media locations, and nighttime images [39,41,42]. We note that the head/tail breaks method clusters the cities based on their size. However, the metropolitan area emphasizes the internal connections between cities. Moreover, the head/tail breaks method does not consider the spatial contiguity of the delineated metropolitan area. Therefore, the head/tail breaks method might not be a good choice for delineating metropolitan areas.

The delineation of metropolitan areas can be conceptualized as a spatial optimization problem, aiming to identify the most beneficial metropolitan boundary while adhering to various constraints. Spatial optimization has long held a significant position within the field of geography. Such optimization techniques can recommend the most advantageous spatial configuration or distribution of study subjects, and they illuminate the implications of specific spatial patterns [43]. The criteria applied in optimization are adaptable based on the distinct challenges at hand, rendering it especially valuable in contexts that demand adaptable planning. This includes areas such as location modeling [44], retail geography [45], political geography [46], and various other domains both inside and outside the field of geography.

The crux of using spatial optimization to delineate metropolitan areas lies in accurately specifying the contiguity constraint, a topic first broached by [47]. Subsequent studies have explored this in the context of delineating CBSAs [48]. While these methods ensure spatial continuity of all included counties, they sometimes result in certain connected counties, not assigned to the metropolitan area, being entirely surrounded by counties that are included in the metropolitan area. As a result, the delineation might contain undesirable "holes", as illustrated in Figure 1. Given that CBSAs in the U.S. generally contain fewer counties and are less intricate in shape than in other countries, such as China, this issue might not manifest during the delineation of CBSAs. However, in our study focusing on China, with its more intricate domestic administrative boundaries, such holes are readily apparent, which is an unresolved issue in the delineation of large regions based on spatial optimization.

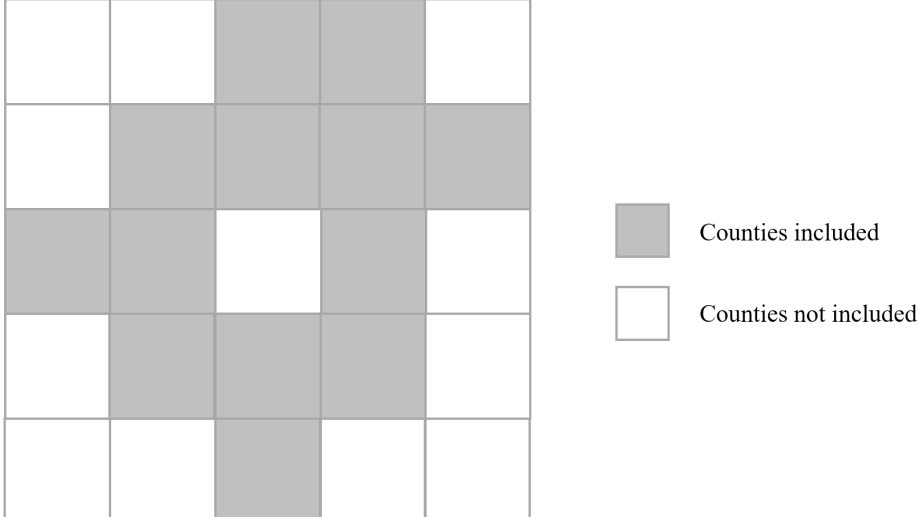

**Figure 1.** Unusual holes in the optimization result.

This research proposed a new spatial optimization model for the delineation of metropolitan areas. The novelty of our work lies in two aspects. Instead of the core-based connection method, our proposed method can measure the strength of all intercounty connections. Moreover, we added new constraints to the spatial optimization model to prevent holes in the optimization results, which makes it more suitable in larger regions such as China. The remainder of this paper is organized as follows: In Section 2, we propose the evaluation method of intercounty connection strength and a spatial optimization model for delineation. In Section 3, we apply our model to delineate the Nanjing metropolitan area and the Lhasa metropolitan area. The discussion and conclusion are provided in Sections 4 and 5.

## 2. Methods

In this section, we introduce a spatial optimization model for delineating metropolitan areas. This method consists of two parts: data processing and model specification. In the data processing part, we determine the study area by calculating the commuting time between the core county and other counties, and the intercounty connection strength of industry and daily life is measured using industry data and chain store data. In the model specification part, a mixed-integer programming model is detailed for delineating metropolitan areas. It aims to maximize all intercounty connections of industry and daily life with constraints in area, number, and spatial contiguity. In particular, we highlight the contiguity constraints of counties that are not assigned to the metropolitan area to prevent holes in the results. Our workflow of the metropolitan area delineation method is shown in Figure 2.

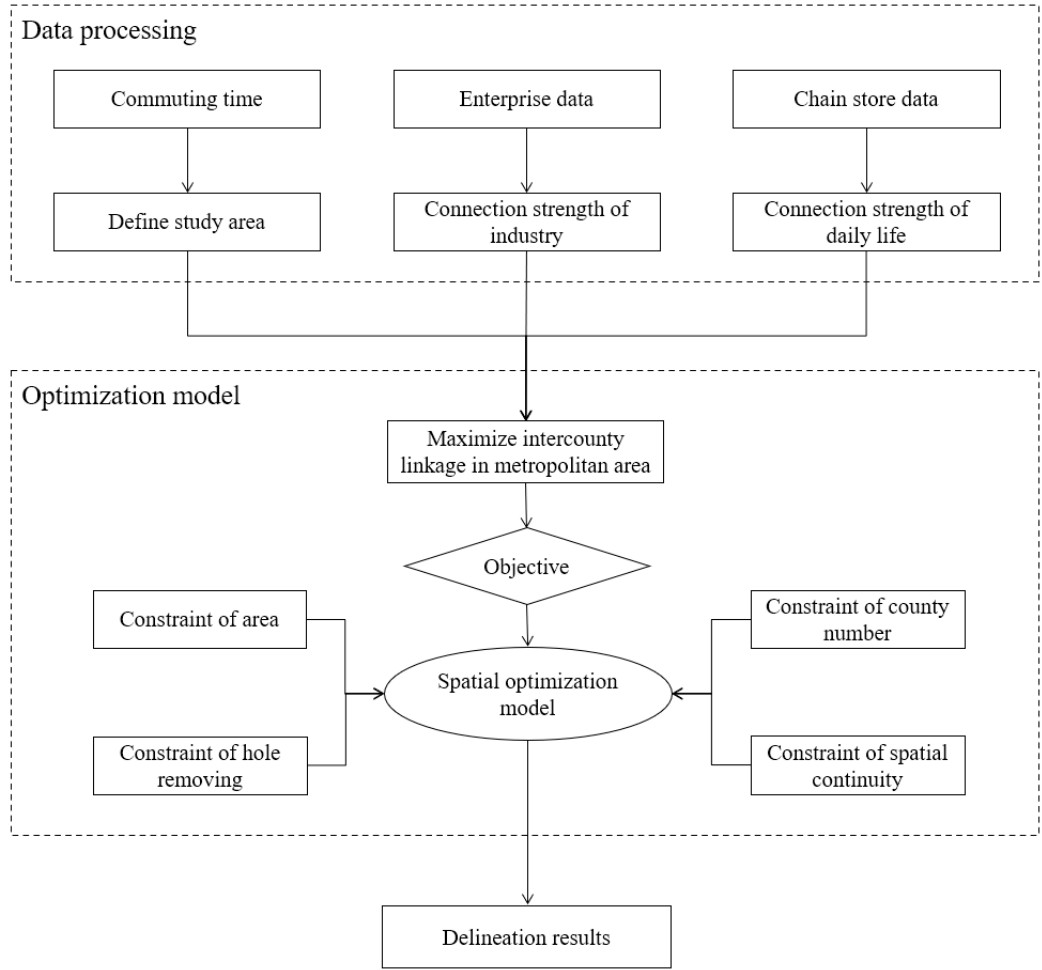

**Figure 2.** Workflow of the metropolitan area delineation method.

*2.1. Connection Strength Measurement*

2.1.1. Measurement of Commuting Time

Internal connections within metropolitan areas largely depend on commuting time, because the main way of connecting core cities with outlying counties is through transportation. Commuting areas reflect the most-essential characteristics of metropolitan areas. All counties in the metropolitan area are within the commuting range of daily life, and through transportation network connections, the integration of municipal infrastructure is achieved [49]. In this study, the travel cost method in grid analysis is used to measure the commuting time from outlying counties to the central city of the metropolitan area.

2.1.2. Measurement of Industry Connection Strength Based on Headquarters and Branch Distribution

Bailly (1995) believed that the headquarters of advanced producer services are most likely to concentrate in world cities or regional central cities, and their branches can often be found in middle-sized or small cities, which show a significant network structure [50]. That is, both the central counties and outlying counties of the metropolitan area are formed under the agglomeration and diffusion of industries. Therefore, regional urban financial flow can be simulated through the number of headquarters and branches of enterprises between different cities. Compared with the return and investment model, which is commonly used in studies on regional economic structure, the measurement of headquarters and branch distribution only needs to collect data on the location of the headquarters and branches, and it does not rely on complex summaries by government statistical departments.

The connection strength of industry between two counties can be calculated as follows:

$$F_{ij} = V_{ij}V_{ji} \tag{1}$$

where $F_{ij}$ is the connection strength of industry between counties $i$ and $j$, $V_{ij}$ is the number of enterprises whose headquarters are in county $i$ and branch in county $j$, and $V_{ji}$ is the number of enterprises whose branch is in county $i$ and headquarters in county $j$.

### 2.1.3. Measurement of Daily Life Connection Strength Based on Chain Store Distribution

Commerce is one of the most important functions of cities, and the chain store, as the mainstream of retail development in the world today, has become a symbol of modern commerce. In 2022, the sales of China's top 100 chain stores reached CNY 1.94 trillion, with nearly 210,000 stores. Chain stores have established themselves as part of our life. The layout of chain stores needs to be considered from various factors, such as transportation, information, capital, raw materials, talent, and sales to achieve optimal regional market control. Hence, the number of brand chain stores in different regions can reflect the connection between residents' lives to some extent.

The connection strength of daily life between two counties can be calculated as follows:

$$L_{ij} = \sum_{k=1}^{m} E_{ik}E_{jk} \tag{2}$$

where $L_{ij}$ is the connection strength of daily life between counties $i$ and $j$, $m$ is the total number of chain stores that we collect, $E_{ik}$ is the number of chain stores $k$ in county $i$, and $E_{jk}$ is the number of chain stores $k$ in county $j$.

### 2.2. Model Specification

We formulate the metropolitan area delineation problem as a mixed-integer programming model with the objective of maximizing the overall intercounty connection strength.

Consider the following notations:
$n$: total number of counties in the study area
$N$: max number of counties in metropolitan area
$A$: max land area of metropolitan area
$P_i$: set of adjacent counties for county $i$
$F_{ij}$: connection strength of industry between counties $i$ and $j$
$E_{ij}$: connection strength of daily life between counties $i$ and $j$
$a_i$: land area of county $i$
$r$: central county of the metropolitan area
$v$: virtual county added outside the study area

The decision variables include:
$y_{ij}$: artificial flow from unit $i$ to $j$ in included counties
$y'_{ij}$: artificial flow from unit $i$ to $j$ in excluded counties

$$x_i = \begin{cases} 1, & \text{if } i \text{ is assigned to metropolitan area} \\ 0, & \text{otherwise} \end{cases}$$

$$x'_i = \begin{cases} 1, & \text{if } i \text{ is not assigned to metropolitan area} \\ 0, & \text{otherwise} \end{cases}$$

$$x'_i = \begin{cases} 1, & \text{if } i \text{ is not assigned to metropolitan area} \\ 0, & \text{otherwise} \end{cases}$$

Figure 3a shows the connection measurement in our model. Unlike traditional delineation methods (Figure 3b), the connections $E_{ij}$ and $F_{ij}$ we account for do not simply represent the connections between central counties and outlying counties. In the pro-

posed model, we examine the strength of all potential intercounty connections. It can ensure a more accurate delineation in developed metropolitan areas, because the structure of metropolitan areas shows a trend for multicore, which cannot be detected through traditional methods [36].

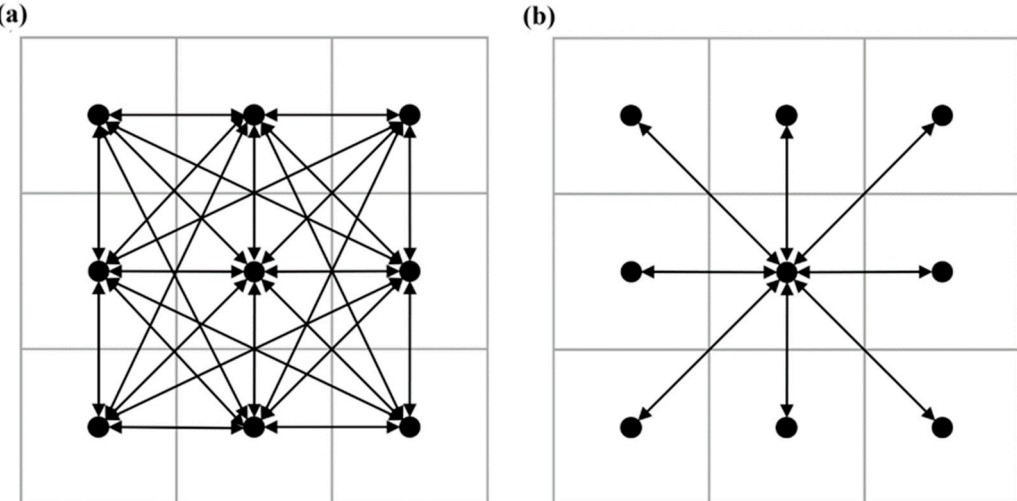

**Figure 3.** Intercounty connection measurement: (**a**) our method; (**b**) traditional method.

The metropolitan area delineation problem may be stated as follows:
Maximize:

$$f_1 = \sum_{i=1}^{n} \sum_{j=1}^{n} t_{ij} F_{ij} \tag{3}$$

$$f_2 = \sum_{i=1}^{n} \sum_{j=1}^{n} t_{ij} E_{ij} \tag{4}$$

Subject to:

$$x_i + x_j \geq 2t_{ij}, \quad \forall i = 1, 2, \ldots, n; j = 1, 2, \ldots, n \tag{5}$$

$$\sum_{i=1}^{n} a_i x_i \leq A \tag{6}$$

$$\sum_{i=1}^{n} x_i \leq N \tag{7}$$

$$\sum_{\{j|j\in P_i\}22} y_{ij} - \sum_{\{j|j\in P_i\}} y_{ji} = x_i, \quad \forall i = 1, 2, \ldots, n; i \neq r \tag{8}$$

$$\sum_{\{j|j\in P_i\}} y_{ji} \leq (N-2)x_i, \quad \forall i = 1, 2, \ldots, n; i \neq r \tag{9}$$

$$\sum_{\{j|j\in P_i\}} y_{jr} \leq (N-1) \tag{10}$$

$$x_r = 1 \tag{11}$$

$$\sum_{\{j|j\in P_i\}} y'_{ij} - \sum_{\{j|j\in P_i\}} y'_{ji} = x'_i, \quad \forall i = 1, 2, \ldots, n; i \neq v \tag{12}$$

$$\sum_{\{j|j\in P_i\}} y'_{ji} \leq (N-2)x'_i, \quad \forall i = 1, 2, \ldots, n; i \neq v \tag{13}$$

$$\sum_{\{j|j \in P_i\}} y'_{jv} \leq (N-1) \tag{14}$$

$$x'_v = 1 \tag{15}$$

$$x_i + x'_i = 1 \tag{16}$$

$$t_{ij} \in \{0, 1\}, \quad \forall i = 1, 2, \ldots, n; j = 1, 2, \ldots, n \tag{17}$$

$$x_i \in \{0, 1\}, \quad \forall i = 1, 2, \ldots, n \tag{18}$$

$$x'_i \in \{0, 1\}, \quad \forall i = 1, 2, \ldots, n \tag{19}$$

$$y_{ij} \geq 0, \quad \forall i = 1, 2, \ldots, n; j \in P_i \tag{20}$$

$$y'_{ij} \geq 0, \quad \forall i = 1, 2, \ldots, n; j \in P_i \tag{21}$$

Objective functions (3) and (4) is to maximize the overall intercounty connection strength of industry and daily life within metropolitan areas. $w_1$ and $w_2$ are the weights of the two objectives with $w_1 + w_2 = 1$, and the overall objective function is specified in Equation (22). According to the specific needs in different planning scenarios, it is possible to adjust the weights:

$$f = w_1 f_1 + w_2 f_2 \tag{22}$$

Constraint (5) specifies that $t_{ij}$ is 1 if and only if $x_i$ and $x_j$ are both equal to 1. In other words, the connection strength between counties $i$ and $j$ will be counted if and only if both counties are within the metropolitan area. Constraints (6) and (7) set the maximum land area and the maximum number of counties within the metropolitan area. More land area and a higher number of counties lead to additional spatial distance between counties. Therefore, improper land area and the number of counties may lead to more commuting time and less spatial connection within the metropolitan area because the volume of spatial interaction decreases as distance separation increases [24]. In this study, the land area and the number of counties constraints refer to relevant government policies implemented in our study areas.

Constraints (8) to (10) ensure that the counties included in a metropolitan area are contiguous. Here, we applied Shirabe's flow model to ensure the contiguity of the delineated metropolitan area [50]. In particular, Constraint (8) adds one unit flow into the overall outflow from county $i$ if it is included in a metropolitan area as an outlying county. Constraint (9) ensures that county $i$ has no flow contribution to the metropolitan area if it is not assigned to it. Constraint (10) assigns the flow from all the outlying counties in the metropolitan area to the core county. Figure 4 shows a possible flow network among the counties included in the metropolitan area. The central county in a metropolitan area is denoted as a sink, and all the outlying counties are denoted sources. The number on a sink, a source, and an arc between two nodes indicate the volume of flow. Each source has an outward flow to one of its adjacent counties whose volume is calculated by summing the inward flows and the flow generated by itself. For example, in Figure 4, county A has an outward flow toward its adjacent county B, and the volume is a sum of its own flow and the inward flow from county C and D. Ultimately, all the flow is gathered to the sink. The flow volumes are decision variables in the model; therefore, they are solved during optimization. We note that the flow network might not be unique. The contiguity of the region is guaranteed as long as all the flow can reach the sink in at least one way, indicating that all counties along the flow are direct or indirect neighbors of the sink. However, as shown in Figure 4, such constraints are not violated when a hole appears in the result. Constraint (11) ensures that the central county is in the metropolitan area.

Constraints (12) to (14) ensure no holes in the optimization results. To achieve this goal, we introduce a virtual county outside the study area and the constraints require all excluded counties to be spatially continuous with the added virtual county. Constraints (12) to (14) are similar to Constraints (8) to (10); therefore, Constraints (12) to (14) ensure that there will be another contiguous region whose sink is the added virtual county, as shown in

Figure 5. If there is a hole in the selected metropolitan area, Constraint (12) will be violated because the flow of excluded counties surrounded by those selected cannot reach the sink. Constraint (15) ensures that the added virtual county is not in the metropolitan area.

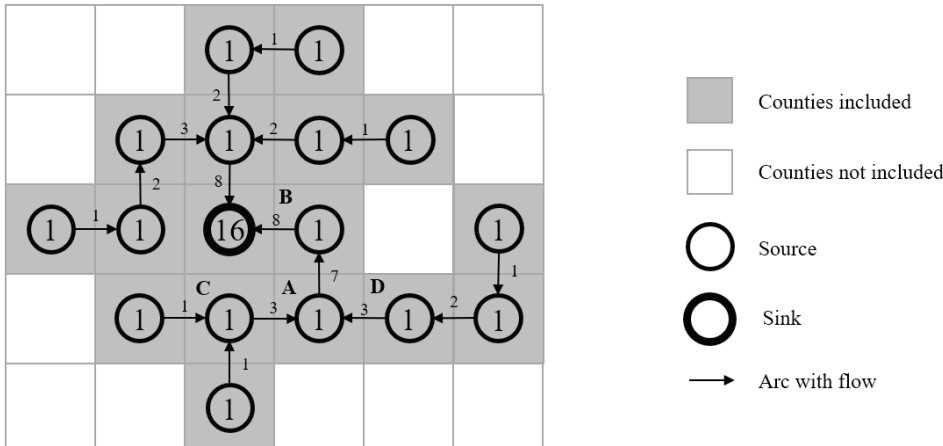

**Figure 4.** A possible flow network in delineating contiguous regions.

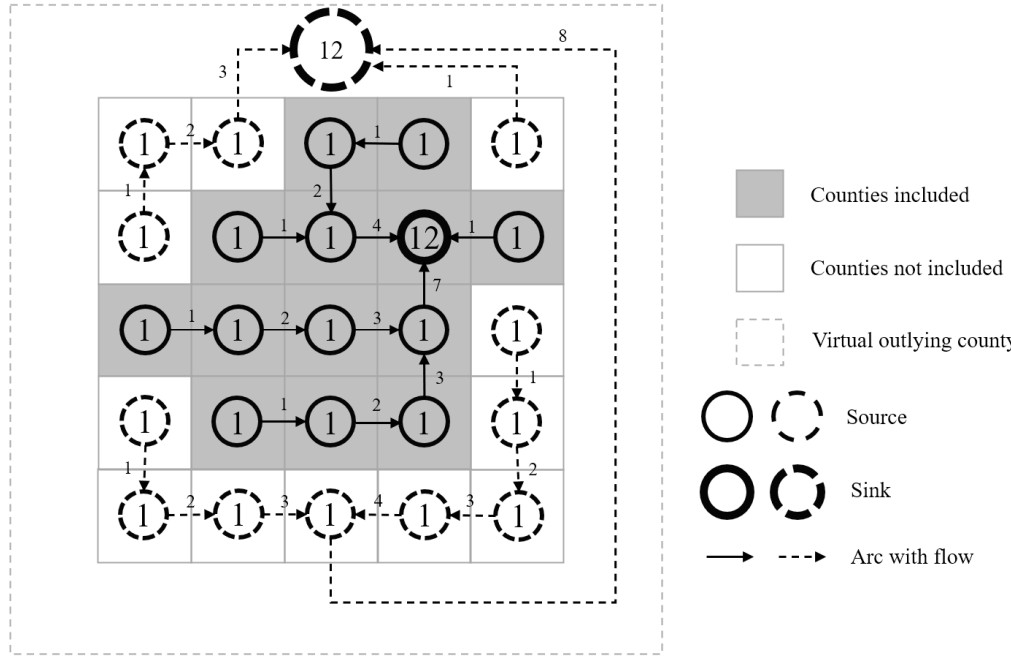

**Figure 5.** An illustration of contiguous metropolitan area without holes.

Constraint (16) ensures that each county is assigned and only assigned to one of the contiguous regions. Constraints (17) to (21) specify that $t_{ij}$, $x_i$, and $x'_i$ are binary decision variables, and $y_{ij}$ and $y'_{ij}$ are nonnegative real numbers.

## 2.3. Model Performance Analysis

We tested our model using a simulated data set. There are four square grid data in the data set, and each of them has a different number of cells, from 16 (4 × 4) to 49 (7 × 7), representing the counties in metropolitan area delineation. The connections between counties are randomly generated a float number from 0 to 1. The test was formulated and solved using the commercial optimization software, IBM ILOG CPLEX 22.1. The computations were performed on an AMD Ryzen 3.0 GHz personal computer equipped with 16 GB RAM.

The test results are shown in Table 1. The $n$ column is the number of grids. The $f$ column is the objective value in delineation results. The N column is the maximum number of counties included in the metropolitan area. The computational time (s) column records the average computational time for solving our model. The solving time limit is set to 1 h. The optimality gap (%) represents the difference between a best-known solution and a value that bounds the best possible solution. Our model performs well for small problems. When the number of spatial units is smaller than or equal to 36, our model could be optimally solved within 6 min. However, when the scale increases to $7 \times 7$, the number of decision variables reaches 10,100, and 5280 constraints are included in the model. Consequently, the model could not be solved optimally within the given timeframe. Given that the number of candidate counties in the metropolitan area delineation is usually not a large number, the proposed model is applicable in this specific application.

**Table 1.** Computational results in model performance test.

| $n$ | $N$ | $f$ | Computational Time (s) | Optimality Gap (%) |
|-----|-----|-----|-----------------------|--------------------|
| 16 | 4 | 9.98 | 0.11 | 0 |
| | 8 | 38.5 | 0.32 | 0 |
| | 12 | 77.2 | 0.09 | 0 |
| 25 | 6 | 21.3 | 0.28 | 0 |
| | 12 | 80.3 | 3.03 | 0.003 |
| | 18 | 168.5 | 2.00 | 0 |
| 36 | 9 | 46.6 | 17 | 0.006 |
| | 18 | 174.3 | 323 | 0.009 |
| | 27 | 386.1 | 81 | 0.009 |
| 49 | 12 | 82.2 | >3600 | 14 |
| | 24 | 309.3 | >3600 | 35 |
| | 36 | 671.8 | >3600 | 8.4 |

## 3. Applications

In this section, we apply the proposed model to delineate boundaries of the Nanjing and Lhasa metropolitan areas. Nanjing is one of the major cities in China, and the Nanjing metropolitan area has been developing for more than 20 years. Although Nanjing has a relatively high urbanized population in China, the city is still undergoing rapid urbanization in terms of net migration, making Nanjing different from many other western cities where suburbanization has been taking place over decades [51]. Therefore, the scope of Nanjing metropolitan area may experience variation since official delineation was over 20 years ago. In contrast, Lhasa is China's most underdeveloped provincial capital. However, it has been under rapid development and urbanization activated by the national policies of Developing West China and Go West [52]. The increasing investments in major infrastructure projects for external communication, transportation, and accommodation have opened up whole new vistas for the city. Nonetheless, Lhasa is still in the infancy stage of metropolitan area construction. Consequently, the Nanjing metropolitan area could be polycentric, while Lhasa may be the only core in the metropolitan area structure. We will show that our model applies to metropolitan areas at both the developed and developing stages. Meanwhile, most metropolitan area studies in China are conducted in first-tier cities such as Beijing or Shanghai, and Nanjing and Lhasa are less familiar to international readers. Constraints include the number of counties, total area, and spatial continuity, with the objective being the maximization of intercounty connection strength within the metropolitan area. We identify the optimal metropolitan area delineation, and based on these findings, we offer planning recommendations to foster the high-quality growth of the metropolitan area. All instances were formulated and solved using the same software and personal computer used in the previous test. Subsequently, the results were imported into ArcGIS 10.7 for visualization and in-depth analysis.

*3.1. Delineation of the Nanjing Metropolitan Area*

3.1.1. Study Area

In January 2003, the "Nanjing Metropolitan Area Plan (2002–2020)" received approval, signifying the formal inception of the Nanjing metropolitan area. Situated in the lower reaches of the Yangtze River, the Nanjing metropolitan area serves as a pivotal nexus bridging the eastern and central regions, as well as the Yangtze and Huaihe Rivers. It amalgamates roles in politics, education, culture, industry, and finance, establishing itself as a central hub for comprehensive transportation within the Yangtze River Delta. The area stands as a crucial pivot in the nation's strategy for opening up and acts as a vital conduit for Anhui Province to integrate with the Yangtze River Delta and partake in its cohesive development. Essentially, it occupies a strategic position in the national regional development blueprint.

As per the "Development Plan for Nanjing Metropolitan Area," this region encompasses Nanjing at its core and extends to closely interconnected neighboring counties. It stretches across two primary provinces, Jiangsu and Anhui, and comprises Nanjing along with 22 other counties, covering a vast expanse of 27,000 km$^2$. By the close of 2022, it boasted a permanent population of approximately 20 million, and its per capita GDP had surged to CNY 140,000.

However, the existing delineation was introduced more than 20 years ago. China has experienced a vast expansion in economy since the 2000s. The scope of the Nanjing metropolitan area needs to be reconsidered to meet regional development requirements. Using a commuting time of 1 h as a criterion, we have defined the study area, as depicted in Figure 6. This area encompasses all counties within the officially delineated Nanjing metropolitan area, in addition to 11 other counties. Collectively, they span a vast 38,000 km$^2$. As of 2021, this region had a GDP of CNY 3.5 trillion and a permanent population nearing 26 million.

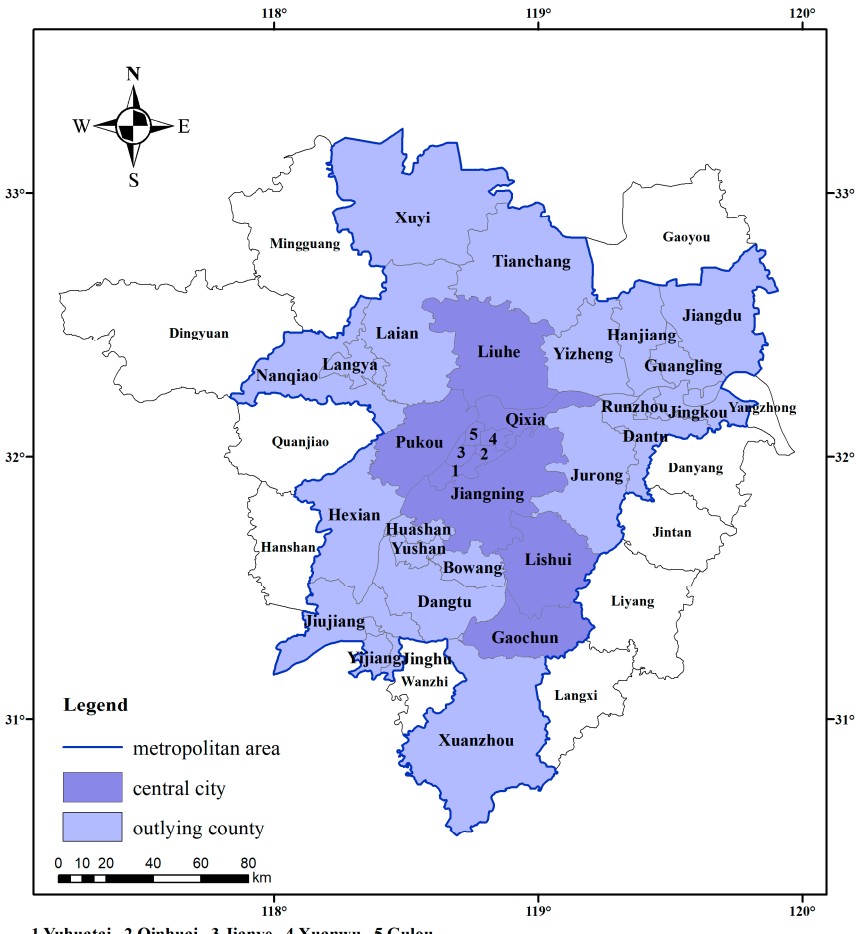

**Figure 6.** Nanjing metropolitan area.

### 3.1.2. Intercounty Connection Strength

Connection strengths are calculated between all pairs of counties within the study area, resulting in the distribution map presented in Figures 7 and 8. The entire metropolitan expanse exhibits a developmental pattern with the Yangtze River as its central axis and Nanjing as its nucleus. This layout indicates a multicore trend. Beyond Nanjing, there is an emergent subcore situated in the northeastern quadrant of the metropolitan area.

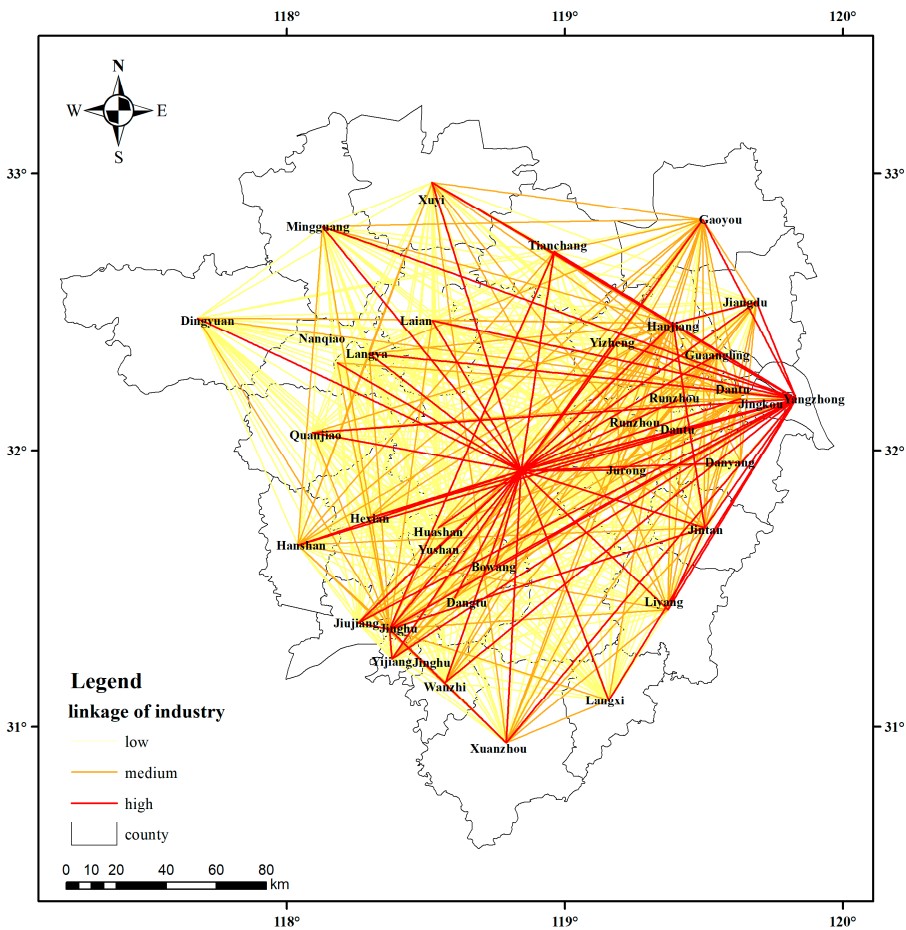

**Figure 7.** Connection strength of industry within the Nanjing metropolitan area.

The connection strength between counties reveals marked disparities. The northeastern counties exhibit strong connections, while those in the southwest demonstrate weaker ties. This dichotomy aligns with the respective economic development levels of these two regions. Nanjing consistently displays strong connections, both industrially and in daily life, with other counties. This underscores Nanjing's pivotal role as the central city in propelling the development of the entire metropolitan area.

Beyond Nanjing, Yangzhong City boasts the highest industrial connection strength, whereas Bowang District lags with the weakest. In terms of daily life connections, Jiangdu District leads, while Wanzhi District trails at the bottom. Of the 1089 connections excluding the central city, 45.9% surpass the average in industrial connection strength, and 37.4% exceed the average in daily life connection strength. This suggests that the industrial connection strength between counties proximate to Nanjing is elevated and relatively even, whereas daily life connection strength exhibits a more polarized distribution.

When evaluating both types of connections, Hanjiang District, Danyang City, and Jiangdu District emerge as the three counties with the strongest connections between other counties. On the other hand, Bowang District, Wanyi District, and Yijiang District have the weakest ties. The network structure of connection strengths shows sparser connections in the west and denser connections in the east, with notable disparities between counties. The

northern region adjacent to Nanjing exhibits a high density, signifying robust industrial and daily life connections. This suggests a marked trend toward urban integration and a distinct clustering phenomenon. Specifically, in the realm of industrial connections, Hanjiang District and Danyang City exert a significant radiating influence, while Yangzhong City stands out in daily life connections. Conversely, the western and southern regions of the study area, barring their ties with Nanjing, display weaker connections with other counties.

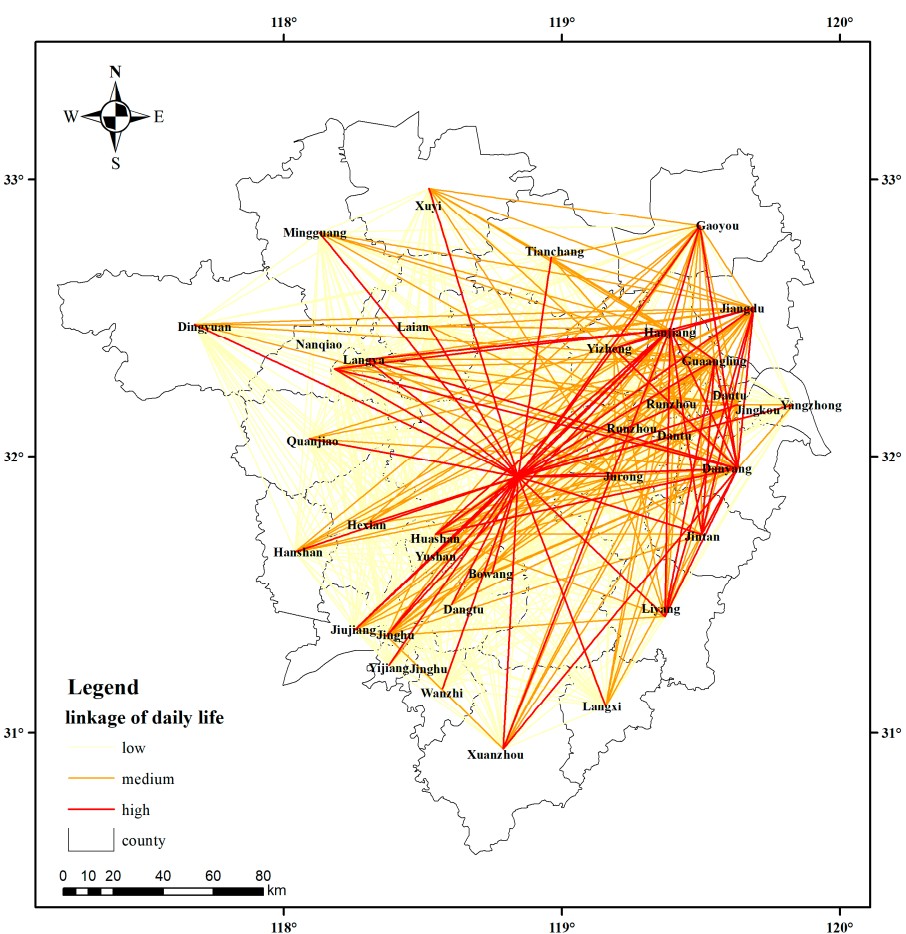

**Figure 8.** Connection strength of daily life within the Nanjing metropolitan area.

### 3.1.3. Metropolitan Area Delineation Result

We delineated the Nanjing metropolitan area with increasing the number of counties in the spatial optimization model, and the results are shown in Figure 9 and Table 2. In general, the Nanjing metropolitan area steadily expands with the increase in the number of counties. Starting from Nanjing and the eastern counties, it will gradually absorb counties in the west and south. The overall connection strength shows an upward trend, while the average connection strength shows a downward trend. Compared to the current delineation, our method preferentially assigns counties with higher levels of development in the eastern study area. In Table 2, the $w_1$ column is the weight of industry connection strength, and the $w_2$ column is the weight of daily life connection strength. The $N$ column is the number of counties included in the metropolitan area. The columns $f_1$, $f_2$, and $f$ are the objective values of industry connection strength, the objective values of daily life connection strength, and the total objective values in delineation results. The computational time (s) column records the average computational time for solving our model. The optimality gap (%) represents the difference between a best-known solution and a value that bounds the best possible solution. The change of total and average intercounty connection strength with the increase in the number of counties is shown in Figure 10.

**Table 2.** Computational results for the Nanjing metropolitan area using different weights.

| $w_1$ | $w_2$ | $N$ | $f_1$ | $f_2$ | $f$ | Computational Time (s) | Optimality Gap (%) |
|---|---|---|---|---|---|---|---|
| 1 | 0 | 20 | 1937 | 0 | 1937 | 3.67 | 0 |
| | | 21 | 2067 | 0 | 2067 | 3.71 | 0 |
| | | 22 | 2197 | 0 | 2197 | 3.74 | <0.01 |
| | | 23 | 2338 | 0 | 2338 | 3.74 | <0.01 |
| | | 24 | 2468 | 0 | 2468 | 3.79 | <0.01 |
| | | 25 | 2596 | 0 | 2596 | 3.79 | <0.01 |
| | | 26 | 2724 | 0 | 2724 | 3.81 | <0.01 |
| | | 27 | 2853 | 0 | 2853 | 3.82 | <0.01 |
| 0.75 | 0.25 | 20 | 1439 | 628 | 2067 | 3.73 | <0.01 |
| | | 21 | 1549 | 648 | 2197 | 3.81 | 0 |
| | | 22 | 1635 | 691 | 2326 | 3.84 | <0.01 |
| | | 23 | 1752 | 711 | 2463 | 3.79 | 0 |
| | | 24 | 1850 | 733 | 2583 | 3.84 | 0 |
| | | 25 | 1946 | 757 | 2703 | 3.92 | 0 |
| | | 26 | 2041 | 784 | 2825 | 3.95 | 0 |
| | | 27 | 2115 | 831 | 2946 | 3.98 | 0 |
| 0.5 | 0.5 | 20 | 959 | 1257 | 2216 | 3.89 | 0 |
| | | 21 | 1003 | 1342 | 2345 | 3.78 | <0.01 |
| | | 22 | 1068 | 1415 | 2483 | 3.83 | <0.01 |
| | | 23 | 1113 | 1505 | 2618 | 3.72 | 0 |
| | | 24 | 1192 | 1546 | 2738 | 3.91 | 0 |
| | | 25 | 1247 | 1612 | 2859 | 3.92 | 0 |
| | | 26 | 1317 | 1662 | 2979 | 3.89 | 0 |
| | | 27 | 1380 | 1719 | 3099 | 3.98 | 0 |
| 0.25 | 0.75 | 20 | 454 | 1932 | 2386 | 3.74 | 0 |
| | | 21 | 480 | 2048 | 2528 | 3.69 | 0 |
| | | 22 | 530 | 2134 | 2664 | 3.77 | 0 |
| | | 23 | 556 | 2257 | 2813 | 3.63 | 0 |
| | | 24 | 583 | 2354 | 2937 | 3.75 | 0 |
| | | 25 | 613 | 2437 | 3050 | 3.69 | 0 |
| | | 26 | 648 | 2513 | 3161 | 3.96 | 0 |
| | | 27 | 680 | 2591 | 3271 | 4.32 | 0 |
| 0 | 1 | 20 | 0 | 2583 | 2583 | 3.74 | <0.01 |
| | | 21 | 0 | 2730 | 2730 | 3.74 | 0 |
| | | 22 | 0 | 2856 | 2856 | 3.75 | 0 |
| | | 23 | 0 | 3009 | 3009 | 3.69 | 0 |
| | | 24 | 0 | 3139 | 3139 | 3.76 | 0 |
| | | 25 | 0 | 3249 | 3249 | 3.76 | 0 |
| | | 26 | 0 | 3352 | 3352 | 3.79 | 0 |
| | | 27 | 0 | 3460 | 3460 | 3.80 | 0 |

With the increase in the number of counties from 20 to 21, Huashan District, previously part of the metropolitan area, was excluded. In place, Langya District and Nanqiao District were incorporated into the delineation. This shift occurred because the combined connection strength of Langya District and Nanqiao District surpassed the aggregate connection strength of Huashan District with any other county within the metropolitan area.

Both Langya District and Nanqiao District are part of Chuzhou City in Anhui Province. They emerged when the original county-level Chuzhou City was bifurcated into two distinct districts. Langya District represents the central urban area of Chuzhou, while Nanqiao District, situated externally to Langya District, shares a border with Pukou District in Nanjing. When compared to Huashan District, their connection strengths with counties in the Nanjing metropolitan area are roughly on par. However, a robust foundational connection exists between Langya District and Nanqiao District. Consequently, as the county count expanded, both districts were concurrently integrated into the metropolitan area. This nuanced shift in the included counties underscores the spatial optimization

model's superiority over other methodologies, because it comprehensively evaluates all potential intercounty connections.

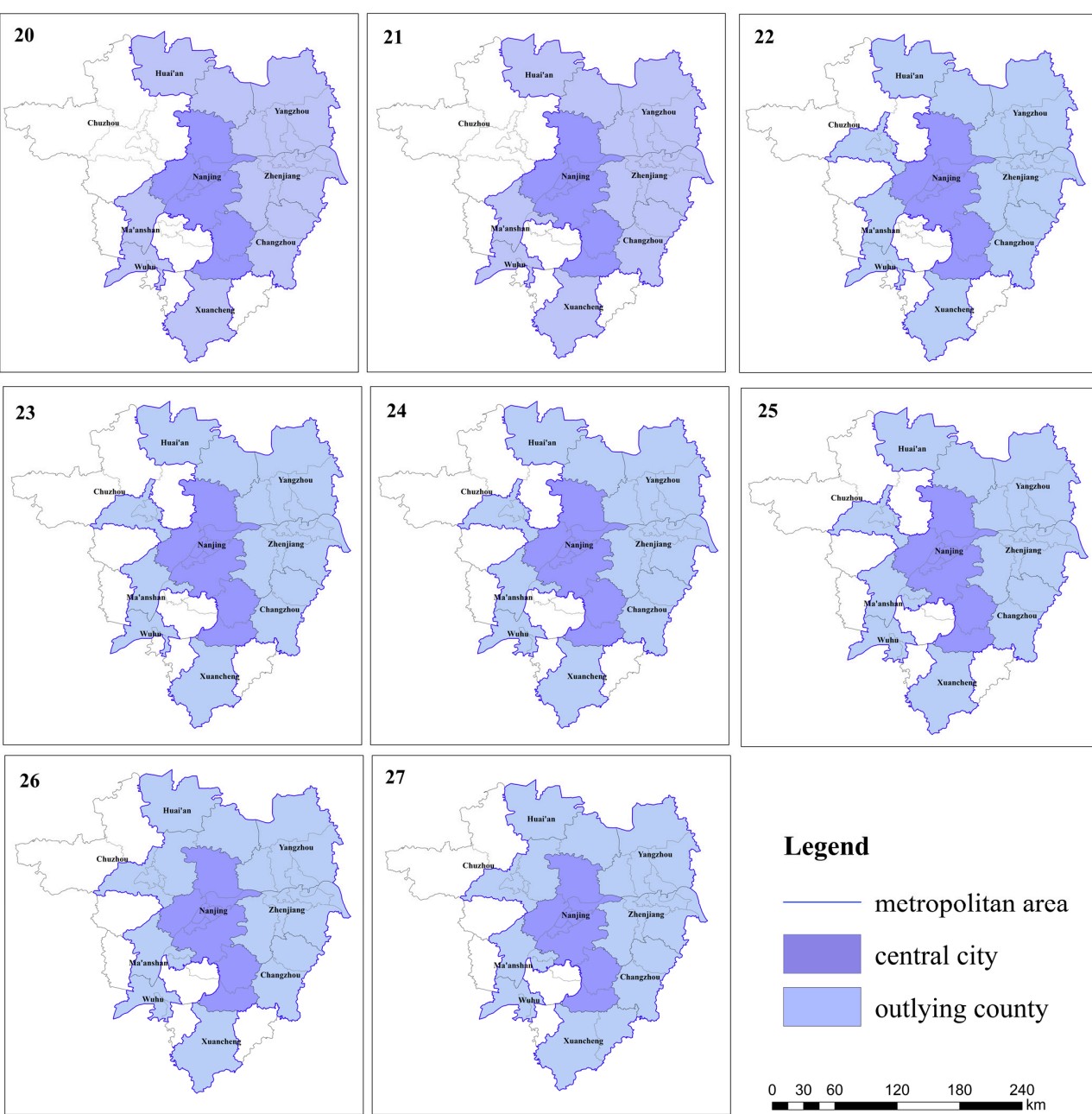

**Figure 9.** Delineation of the Nanjing metropolitan area ($w_1 = 0.5$, $w_2 = 0.5$).

When the county count is set at 23, matching the current number of counties in the Nanjing metropolitan area, a comparison with the existing delineation of the Nanjing metropolitan area was made. While the results largely align, our delineation includes Danyang City, Yangzhong City, Gaoyou City, Jintan District, and Liyang City, replacing Yijiang District, He County, Yushan District, Bowang District, and Laian County. The total land coverage remains consistent at 28,000 km². However, the total GDP in our delineation reaches CNY 3.1 trillion, marking a 10% increase over the existing delineation. Furthermore, the per capita GDP stands at CNY 141,000, a notable 25% surge compared to the current delineation.

We note that the existing delineation emphasizes a harmonized development across the metropolitan area, underscoring the catalytic role of Nanjing. This approach fosters enhanced interactions and exchanges between Jiangsu and Anhui provinces, cultivating

a synergistic development trajectory that leverages mutual strengths. In contrast, our delineation, rooted in spatial optimization, aims to intensify the internal ties and boost regional cohesion primarily within Jiangsu Province. Concurrently, it identifies potential multicore configurations within the study area, offering valuable insights for the potential outward expansion of the Nanjing metropolitan area.

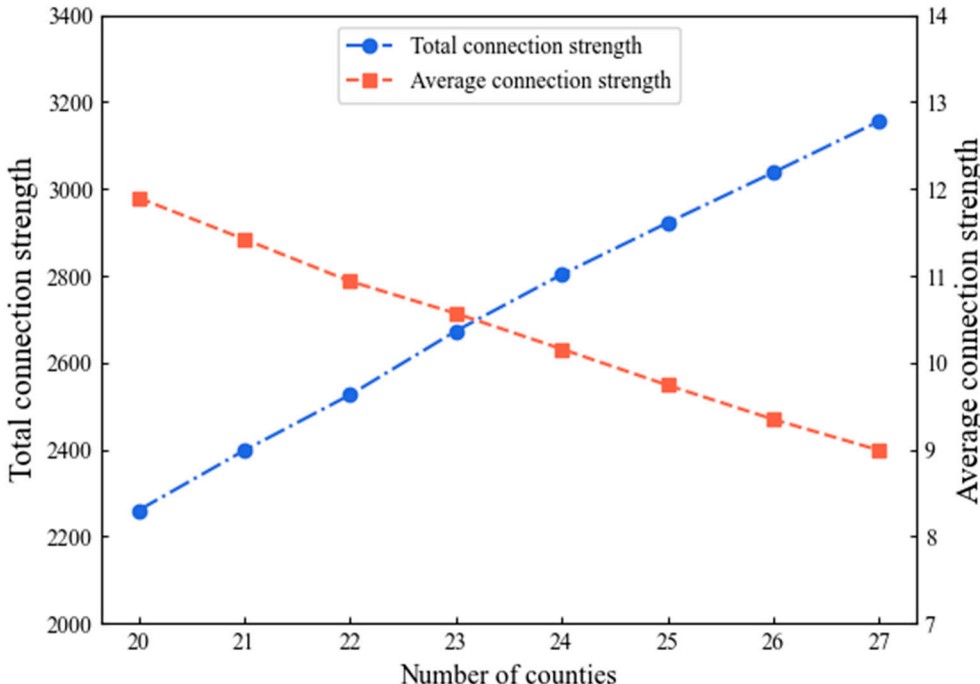

**Figure 10.** Change of total and average connection strength with the increase in the number of counties.

In terms of economics, the five counties added to our delineation exhibit robust economic vigor, with each boasting a per capita GDP exceeding CNY 100,000. Gaoyou City, Danyang City, Jintan District, and Liyang City all have GDPs surpassing CNY 100 billion. Notably, Jintan District's per capita GDP, at CNY 180,000, stands as the highest among all counties in the study area. In the existing delineation, barring Yijiang District, the per capita GDP of the four excluded counties falls below CNY 100,000. Moreover, their collective GDPs are beneath the metropolitan area's average.

In contrast to the official Nanjing metropolitan area, our approach emphasizes the clustering and interplay between counties. As a result, it incorporates eastern counties, which are economically more robust, into the metropolitan area. The existing delineation prioritizes the metropolitan area's radiating influence on its peripheries. It contemplates the southward expansion of the metropolitan area, aiming to bolster the urban growth of both Ma'anshan City and Wuhu City.

From a locational standpoint, our delineation underscores the connections within the eastern region, specifically the internal ties within Jiangsu Province. Gaoyou City, situated adjacent to the Nanjing metropolitan area, can act as a pivotal node for the expansion of the Nanjing metropolitan area toward both Taizhou City and Huai'an City. This positions Gaoyou City as a linchpin, bridging both north–south and east–west connections. Danyang City and Yangzhong City, located to the east of Nanjing and nestled along the Yangtze River, can amplify the influence on counties situated along the river's middle reaches. Jintan District and Liyang City, to the south of Nanjing, occupy a strategic position between Nanjing and Shanghai. This makes them instrumental in facilitating exchanges and interactions between these two major Chinese cities.

In contrast, the existing delineation aims to bolster interprovincial interactions. It incorporates several counties from Anhui Province, including Wuhu City, Ma'anshan City, Chuzhou City, and Xuancheng City, into the Nanjing metropolitan area. This approach

seeks to dissolve administrative boundaries, extend the metropolitan area's influence, and offer exemplary models for the development of cross-provincial metropolitan regions.

The differences between our delineation and existing delineation reveal the imbalanced development in the metropolitan area. The western counties are relatively underdeveloped. Their primary focus should be on economic development and boosting cross-provincial collaboration capabilities. While these counties are geographically proximate to Nanjing, they possess a weaker economic base and fall under the jurisdiction of Anhui Province. Their ties with counties in Jiangsu Province are weak. It is imperative for these counties to not only fortify their economic foundation but also to amplify their interprovincial collaboration with neighboring counties by overcoming administrative obstacles.

When the county count increased to 27, our analysis revealed that almost all counties currently associated with the Nanjing metropolitan area were included, with the notable exception of Bowang District. A deeper examination of industrial and daily life connections indicated that the ties between Bowang District and other counties were markedly weak, placing it second to last among all counties in the study area. This could be attributed to Bowang District's relatively recent establishment in 2012. Its nascent stage of development has led to a slower urbanization pace, limited residential zones, and a diminished capacity to attract businesses and the retail sector.

Figure 11 shows a delineation result without the hole-removing constraints that we added to the spatial optimization model. In contrast to the last delineation result in Figure 9, Yushan District and Bowang District are surrounded by counties included in the Nanjing metropolitan area. An apparent hole appears in the delineation result, which is inappropriate for urban planning. The difference between the two delineation results shows the effectiveness of our added constraints.

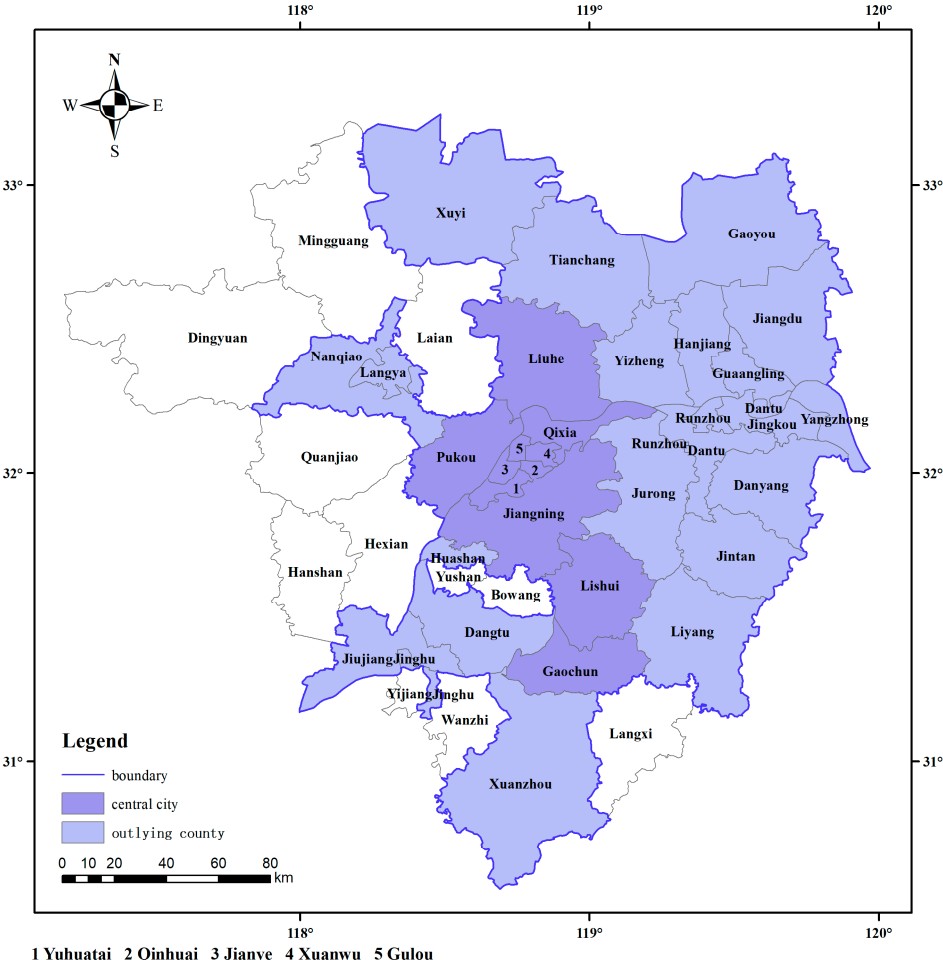

**1 Yuhuatai 2 Qinhuai 3 Jianye 4 Xuanwu 5 Gulou**

**Figure 11.** Delineation result of the Nanjing metropolitan area without hole-removing constraints.

*3.2. Delineation of the Lhasa Metropolitan Area*

3.2.1. Study Area

Lhasa City serves as the capital of the Tibet Autonomous Region (later referred to as Tibet). Tibet, situated in the southwest of the Qinghai–Tibet Plateau, shares borders with countries and regions such as Myanmar, India, Bhutan, and Nepal. Boasting an average elevation exceeding 4000 m and spanning approximately 1.2 million square kilometers, Tibet is China's second-largest province. As of the close of 2021, it had a permanent population of 3.66 million and a GDP of CNY 208 billion, marking a 6.7% growth from the previous year. However, Tibet's urbanization rate stands at 36%, significantly trailing the national average of 64%.

Given its relatively undeveloped nature, the Lhasa metropolitan area has historically received limited focus. However, Tibet is a pivotal region for China's external outreach and serves as a central conduit for China's interactions with South Asia, underscoring its strategic importance. As such, the delineation of a metropolitan area for Lhasa is of paramount significance, which not only aids in its regional planning and accelerated growth but also bolsters its competitive strength.

Considering Tibet's expansive land area and the considerable administrative span of many of its counties, the "Tibet Land Space Plan (2021–2035)" has extended the commuting time threshold to 3 h. The study area, as depicted in Figure 12, encompasses 27 counties in proximity to Lhasa. Collectively, these counties cover an area of 290,000 km², contain a permanent population of 2.14 million, and have a combined GDP of CNY 136.4 billion.

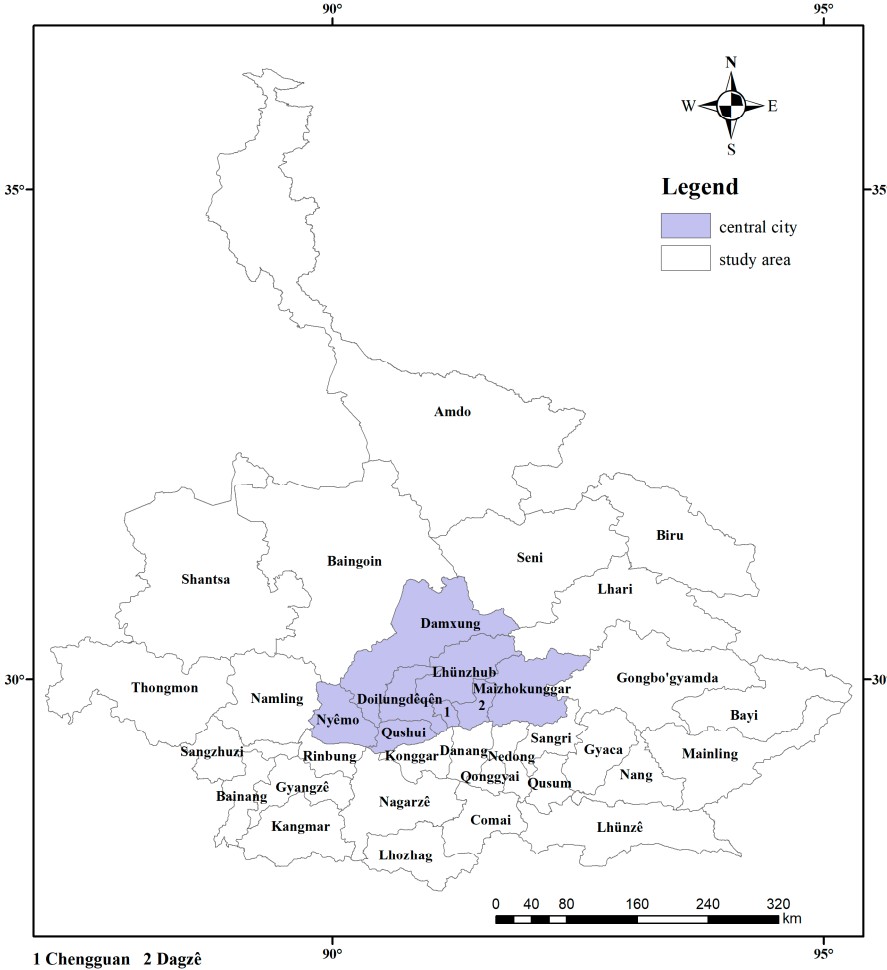

**Figure 12.** The Lhasa metropolitan area.

### 3.2.2. Intercounty Connection Strength

Connection strengths are derived for each pair of counties in the research area as illustrated in Figures 13 and 14. In general, the connections within the study area exhibit a monocentric structure with Lhasa as its core. Notable disparities are evident between counties. The connection strength between Lhasa, the central city, and regions to its northwest and southeast is relatively weak, with its influence on other counties diminishing rapidly. Apart from a select few counties, most exhibit limited interactions, especially those on the periphery of the study area. The entire region's development concentrates along the Brahmaputra River and the Qinghai–Tibet Railway, suggesting a latent potential for the evolution of a metropolitan area in the region.

When compared with the Nanjing metropolitan area, the overall connection strength within the study area surrounding Lhasa is noticeably subdued, with the peak value being three times lower. This underscores the nascent stage of Lhasa's development. The connections between Lhasa and its peripheral counties vary considerably, largely contingent on the geographical location of these outlying counties. Lhasa's influence is pronounced on counties along the Brahmaputra River and the Qinghai–Tibet Railway. However, its impact diminishes for other counties, indicating a need for Lhasa to amplify its catalytic role as the central city.

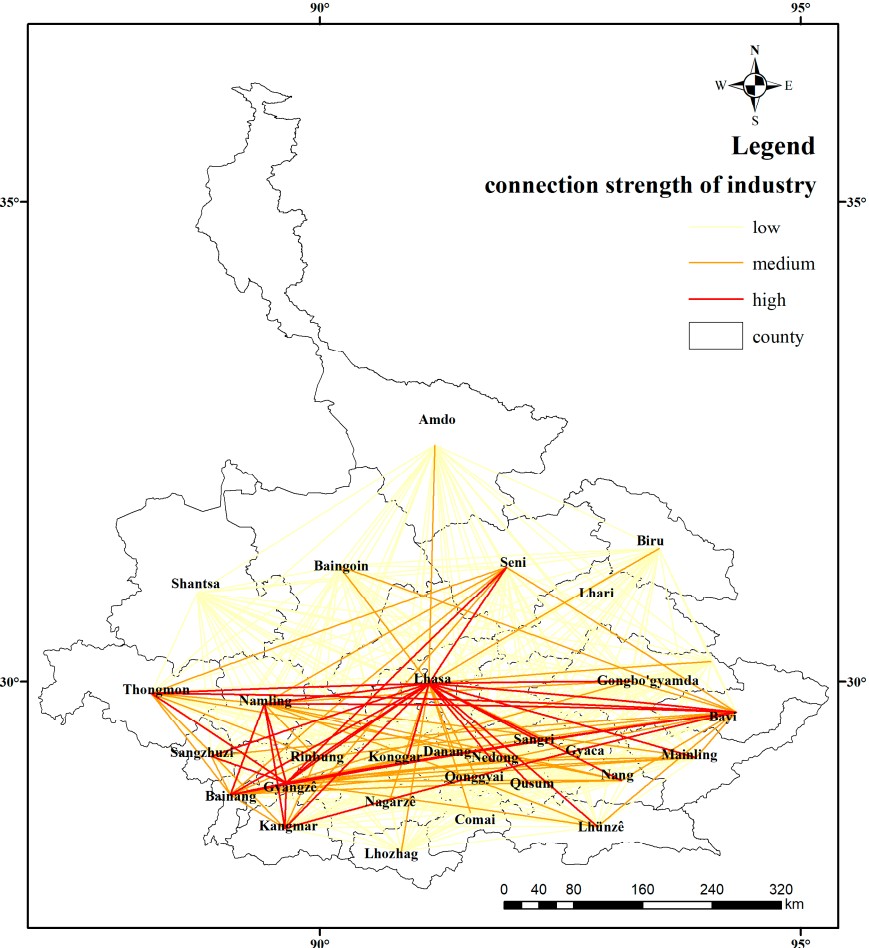

**Figure 13.** Connection strength of industry within the Lhasa metropolitan area.

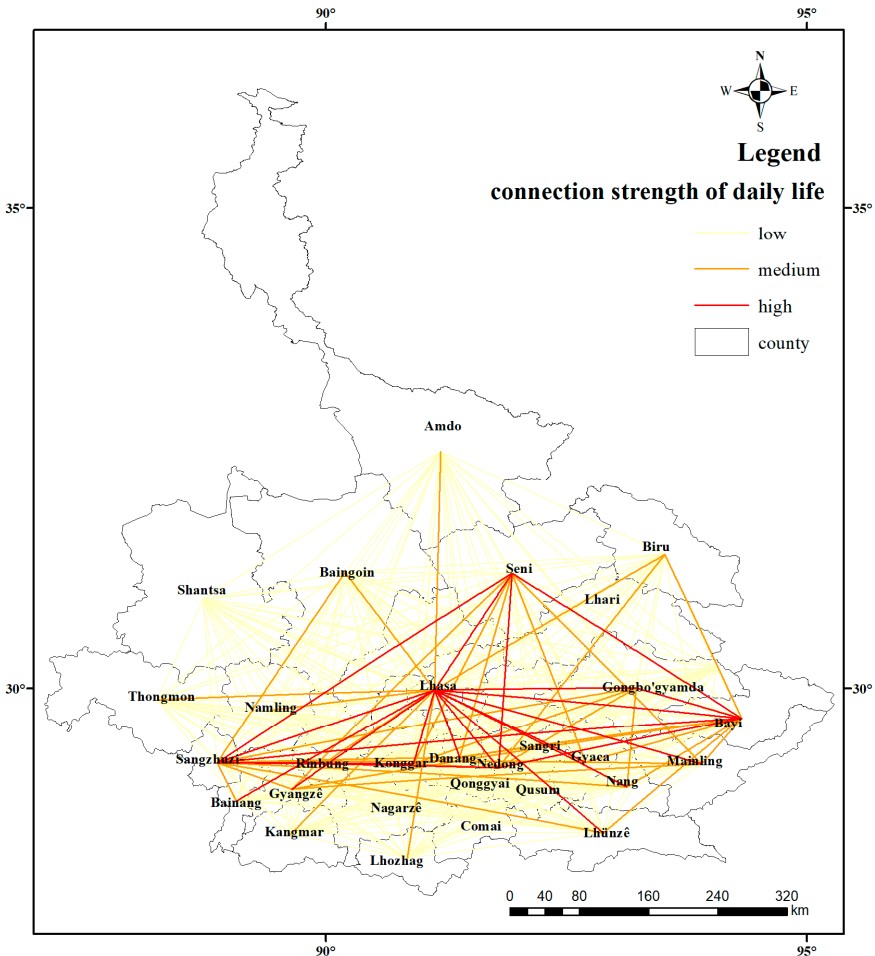

**Figure 14.** Connection strength of daily life within the Lhasa metropolitan area.

### 3.2.3. Metropolitan Area Delineation Result

Figure 15 presents the delineation results of the Lhasa metropolitan area, as derived from our spatial optimization model. Computational results are shown in Table 3, whose columns are similar in definition to Table 2. The Lhasa metropolitan area encompasses 20 counties, spanning a total area of 150,000 km². It houses a population of 1.84 million and boasts a combined GDP of CNY 128 billion, which represents two-thirds of Tibet's entire GDP. The per capita GDP within this metropolitan area stands at CNY 70,000, marking a 20% increase over the average per capita GDP across Tibet.

**Table 3.** Computational results for the Lhasa metropolitan area using different weights.

| $w_1$ | $w_2$ | $f_1$ | $f_2$ | $f$ | Computational Time (s) | Optimality Gap (%) |
|-------|-------|-------|-------|-----|------------------------|---------------------|
| 1     | 0     | 446   | 0     | 446 | 2.68                   | 0                   |
| 0.75  | 0.25  | 335   | 93    | 428 | 2.84                   | 0                   |
| 0.5   | 0.5   | 217   | 195   | 412 | 2.83                   | 0                   |
| 0.25  | 0.75  | 109   | 292   | 401 | 2.73                   | 0                   |
| 0     | 1     | 0     | 390   | 390 | 2.55                   | 0                   |

Overall, while the Lhasa metropolitan area remains in a nascent stage of development, the central city of Lhasa exerts a discernible influence on its surrounding counties. Beyond the central city, there are several clusters of counties with robust internal connections, indicating the potential for further metropolitan development. The delineation aligns with Tibet's urban planning strategy, which positions Lhasa as the core and uses the

Brahmaputra River as a connecting axis. However, adjustments may be necessary to cater to specific developmental needs and objectives.

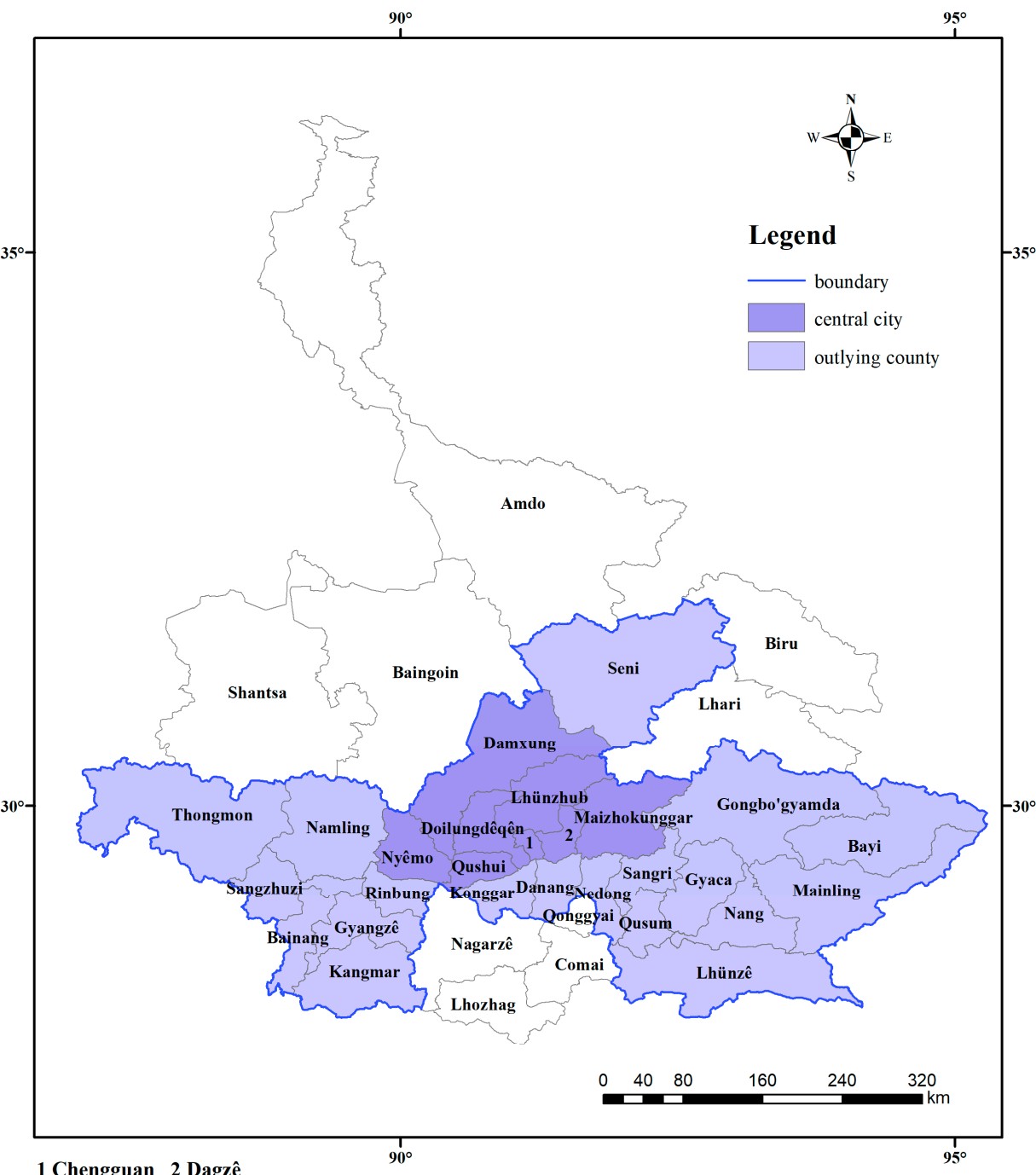

**Figure 15.** Delineation of the Lhasa metropolitan area ($w_1 = 0.5$, $w_2 = 0.5$).

Within the Lhasa metropolitan area, Sangzhuzi District, Bayi District, and Seni District stand out as economic powerhouses, collectively contributing to over 75% of the metropolitan area's GDP when combined with Lhasa. Notably, the Qinghai–Tibet Railway links Seni District and Lhasa, providing a solid foundation for commuting and connectivity. Sangzhuzi District, Bayi District, and Lhasa, on the other hand, are positioned along the Brahmaputra River. The valley of the Brahmaputra River represents the most urbanized and developmentally primed regions in Tibet.

In contrast, the remaining counties struggle with challenges such as limited transportation access, underdeveloped socioeconomic frameworks, and tenuous connections with Lhasa. These observations underscore a prevailing trend in Lhasa's developmental strategy: a focus on outward expansion along the key arteries of the Brahmaputra River and the Qinghai–Tibet Railway. For them, a concerted effort to enhance infrastructure, particularly transportation, is essential. Existing transport links should be enhanced, which will bolster intercounty communication by enhancing accessibility and establishing a commuting circle centered on Lhasa.

When compared with the Nanjing metropolitan area, the Lhasa metropolitan area spans a more extensive land area but is characterized by a smaller population and weaker competitiveness. Furthermore, due to notable developmental disparities among its counties, the Lhasa metropolitan area has not evolved into a multicore urban network that is similar to the Nanjing metropolitan area. Instead, it exhibits a monocentric with a radial connection structure. Compared with Nanjing, Lhasa needs to amplify its influence on peripheral counties and broaden its outreach by scaling up and diversifying its functions. By bolstering socioeconomic vitality, the Lhasa metropolitan area can eventually enhance interactivity within the metropolitan area and foster the emergence of holistic development.

Nevertheless, the Lhasa metropolitan area's developmental potential is anchored in its relatively weaker administrative barriers and the profound cultural coherence among its counties, paving the way for integrated urban development. A cohesive coordination mechanism and development strategy must be instituted to cater to the diverse requirements of different counties. Concurrently, it is crucial to preserve the unique plateau environment and the region's distinct cultural heritage. By systematically bolstering its economy and establishing a comprehensive transportation network, the Lhasa metropolitan area can cultivate a high-quality network of influence, yielding dividends for both China and its neighboring nations.

## 4. Discussion

Compared to conventional delineation methods, our approach offers a nuanced understanding of the metropolitan area's structure by assessing the comprehensive intercounty connections in both industry and daily life. This allows for the identification of a multicore structure, which traditional methods might overlook. Moreover, our method offers a delineation that yields higher benefits and can be tailored to suit developmental needs across different phases by using the spatial optimization model. We applied this methodology to the delineation of both the Nanjing and Lhasa metropolitan areas. When compared with current delineations, our approach appears to prioritize the deepening of connections within the metropolitan area. This focus can swiftly bolster regional cohesion and elevate the metropolitan area's competitiveness, maximizing developmental impact. In addition, we managed to describe the delineation problem with a mixed-integer programming formulation. Therefore, it demonstrated outstanding performance in solving optimal results. The optimal delineation can be solved within seconds.

We also proposed an additional constraint to prevent holes in delineation results. A virtual county outside the study area is introduced, and all the excluded counties are required to be spatially continuous with the added county as the central county. It was utilized in our optimization model and its efficacy was demonstrated through contrast with results before the application. This helps present a more desirable delineation result in countries and regions with numerous counties and intricate administrative boundaries.

However, the purpose of establishing metropolitan areas extends beyond merely amplifying regional competitiveness. Metropolitan areas also play a pivotal role in catalyzing the growth of less-developed regions, fostering integrated urban development, and enhancing the quality of life for residents. As such, urban planning must seek a delicate balance. It is imperative to envision the metropolitan area's long-term trajectory and make necessary adjustments to the delineation results to align with on-ground realities.

Our testing revealed that our proposed model is time-intensive on a large scale and may not yield optimal results. Although most metropolitan areas comprise a few counties with simpler adjacency relationships than those in a grid world, addressing this issue is crucial in our methodology. A potential improvement lies in limiting the number of variables and constraints in the optimization model while still addressing the delineation problem effectively.

Moreover, our model's focus on industry and daily life connections does not fully capture the complexity of urban interactions in real societies, as these are merely components of the broader connectivity within metropolitan areas. Incorporating additional dimensions such as economy, transportation, ecology, and culture could yield more comprehensive delineation results. Additionally, our analysis did not examine the impact of each connection type on regional development nor establish a method to determine the weights of these connections.

Data limitations led us to use headquarters–branch distributions and chain store data to assess intercounty connection strength. We overlooked activity flow data, which more directly reflects real-time connections, such as enterprise investment, freight transport monitoring, phone calls, and internet interactions among residents in different counties.

Furthermore, despite meeting contiguity and hole-removing constraints, the delineating result might still not be desired in shape. For example, Bowang District is not included in the Nanjing metropolitan area due to its weak connections, creating a dent in the delineation result. We note that the shape of the metropolitan area might also be critical. The compactness, which aims for a circle-like shape to maximize accessibility throughout a region, might be an objective of the optimization model. However, developing an explicit measure of compactness remains a significant challenge in the field of spatial optimization [53].

## 5. Conclusions

In this study, we introduce a spatial optimization model for the delineation of metropolitan areas. Traditional approaches predominantly emphasize the interactions between counties within the metropolitan area and its central city, often overlooking the dynamics between peripheral counties. Such methods fall short in recognizing potential secondary cores in multicore metropolitan areas, where interactions between specific peripheral counties might be significant. Our model rectifies this by evaluating all intercounty industrial and daily life connections within the metropolitan domain. It aims to identify delineations that maximize overall interactions. We also apply extra contiguity constraints to prohibit holes in delineation results by utilizing a virtual node. We tested this approach on the Nanjing and Lhasa metropolitan areas, yielding results that align with the developmental aspirations of these regions and enhance their overall competitiveness. Based on these delineations, we have also offered pertinent policy recommendations.

There are a few directions for future studies. First, our methodology employs enterprise and chain store data to gauge intercounty connections. However, these data sets only scratch the surface of the intricate tapestry of human interactions across regions. Consequently, future studies might delve deeper into measuring these connections. Factors such as collaborative innovation, ecological partnerships, historical roots, and cultural affinities could provide richer insights into urban integration. Furthermore, the dynamics within regions influenced by multiple metropolitan areas warrant additional exploration. For example, the Yangtze River Delta region houses several developed metropolitan areas, such as Hangzhou, Hefei, and Shanghai. The fringes of any given metropolitan area in this region are invariably impacted by its neighbors. Crafting a balanced and judicious delineation plan in such contexts is a challenge that merits closer examination.

**Author Contributions:** Conceptualization, Gusiyuan Wang and Wangshu Mu; methodology, Gusiyuan Wang and Wangshu Mu; data curation, Gusiyuan Wang; writing—original draft preparation, Gusiyuan Wang; writing—review & editing, Gusiyuan Wang and Wangshu Mu; visualization, Gusiyuan Wang; supervision, Wangshu Mu; project administration, Wangshu Mu; funding acquisition, Wangshu Mu. All authors have read and agreed to the published version of the manuscript.

**Funding:** This research was funded by the Second Tibetan Plateau Scientific Expedition and Research Program (STEP) (Grant No. 2019QZKK0608), the National Key R&D Program of China (Grant No. 2022YFC3800105), the National Natural Science Foundation of China (Grant No. 42301476), the Strategic Priority Research Program of the Chinese Academy of Sciences (Grant No. XDA23100303), and a Project Supported by the State Key Laboratory of Earth Surface Processes and Resource Ecology (2022-ZD-04).

**Data Availability Statement:** The data presented in this study are available on request from the corresponding author. The data are not publicly available due to privacy concerns.

**Acknowledgments:** We would like to thank the high-performance computing support from the Center for Geodata and Analysis, Faculty of Geographical Science, Beijing Normal University.

**Conflicts of Interest:** The authors declare no conflicts of interest.

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
