# Peer review of "A Spatial Optimization Model for Delineating Metropolitan Areas"

_ijgi, doi:10.3390/ijgi13020051_

Round 1

Reviewer 1 Report

Comments and Suggestions for Authors

This manuscript introduces a compelling work that employs spatial optimization to delineate metropolitan areas using spatial interaction flows. The spatial characterization of urban areas is a pivotal issue in urban research and has received considerable attention; however, only a limited number of studies have approached the problem from an optimization perspective. The proposed algorithm is interesting. The case studies, in Nanjing and Lhasa, are interesting and well explained. Several significant challenges must be addressed to ensure the robustness and validity of the methodology prior to publication.

1. Literature Review. While the authors argue that the optimization method outperforms threshold-based methods, particularly in detecting polycentric urban structures, it is important to acknowledge that numerous methods have been proposed to reveal such structures, including multi-level urban structures in cities (e.g., Cai et al., 2017; Masucci et al., 2018). Additionally, the issue of delineating urban areas based on spatial interaction flows has also been explored. The authors should include a review of these methods in their manuscript.

2. Optimization Formulation.

1) Maximum area constraint. The area constraint is calculated based on the "average metropolitan areas of all cities in China". The approach of using previously delineated urban areas to determine the urban area of the next city may lead to a circular problem. Additionally, the assumption that the urban areas of all cities in China should be comparable may not hold true, given the significant differences in urbanization levels across the country. Furthermore, it is unclear why a constraint on the number of counties and a constraint on the urban area are both needed.

2) Hole filling algorithm utilized in the optimization method. The hole filling algorithm appears to be a significant novelty of the approach; however, the manuscript does not provide a clear and comprehensive explanation of the method, making it difficult to understand its practical implementation and efficacy. For instance, the manuscript does not explain how the “central county” is determined from which the flow starts or how the flow direction is determined when a cell can flow into any of its neighbors. These questions must be addressed to provide a better understanding of the methodology and to validate its usefulness.

3) Scalability of the algorithm. As mentioned in lines 266-267, constraints 12 and 14 could become computationally expensive when implemented on a large spatial area or at high spatial resolutions. This raises questions about the algorithm's ability to handle large-scale datasets efficiently. It would be beneficial to discuss the potential limitations and computational challenges associated with implementing the algorithm on a larger scale, and explore possible strategies for improving its scalability.

It would be also beneficial to improve the manuscript by addressing the following points:

4) The sensitivity test on the number of counties included could be expanded to cover a larger range. This would provide a better demonstration of the algorithm's scalability and its ability to handle larger datasets.

5) Visualizing the marginal additions in spatial interaction intensities with increased urban areas and counties allowed in the urban core would be an interesting addition to the manuscript.

6) Optionally, comparing the delineated urban areas with areas from urban percolation algorithms (Masucci et al., 2018; Zhong et al., 2014), which can be more efficient, may add value to the manuscript. It would also be useful to explore the similarities and differences between the proposed algorithm and community detection algorithms in computer science, which could provide insights into potential future developments of the methodology.

Author Response

Please see the file attached.

Reviewer 2 Report

Comments and Suggestions for Authors

This paper introduced a spatial optimization model for the delineation of metropolitan areas. This model aims to maximize intercounty connection strength in terms of both industry and daily life. The authors applied the model to Nanjing and Lhasa metropolitan areas. The current presentation is very lengthy which reads like a thesis. Please make the below revisions.

1.       Please combine the Introduction and Background, and shorten this section. Please highlight your contributions, and compare the advantages and disadvantages of the previous studies. Please do not present so much existing knowledge.

2.       Line 92: Greater Tokyo Area spans 13555 km2 (km2)

3.       Section 4: Applications. Why do you choose Nanjing and Lhasa metropolitan areas? The current presentation reads very lengthy. Could you please combine the two cities? And especially to compare the differences and similarities.

Author Response

Please see the file attached.

Reviewer 3 Report

Comments and Suggestions for Authors

This a very well presented paper, treating delineation of a metropolitan area as a spatial optimization problem, with the authors providing certain methodological improvement on well established solutions to similar problems.

The structure of the paper is appropriate, the methodology is clearly presented, while the case studies delineating the boundaries of the Nanjing and Lhasa metropolitan areas, are interesting and well justified. 

Moreover, the paper merits publication and fits the aim and scope of the  International Journal of Geo-Information.

However I suggest certain revisions before publications. I classify them as "minor" because the most important one (remark 2) is related to "cutting down" some sections (5.2, 5.3) rather than altering the paper content, methoology etc. This will keep the paper more clear, short and focused on its true innovation.

1. Try to enrich and extend the references cited throughout the paper with more international research, and most importantly researchers and authors for Europe of U.S.A. examining spatial optimization problems, as well as Larger Urban Zone delineation (e.g. Urban Morphological Zones). This is a research issue that goes back to the period of quantitative revlolution in geography, so I propose that certain links are drawn, to better justify how the present work extends or break-through existing research pathways.

2. In the Discussions' section, I propose that only methodological findings resulting from the case study analysis should be discussed. To my view, the whole part of policy implications is rather irrelevant to the research focus of this work. The method presented to define the boundaries is well justified, however, I understand it could be used into very different cases and contexts, probably with adjustments on the variables considered (e.g. strength of daily life might be captured by other variables rathen than chain stores!), the spatial units used in the analysis (e.g. instead of counties using finer resolution units) etc. These are more important points to discuss, and I don't see how the specification of policy implications regarding e.g. regional policies to support metropolitan growth in the 2 case study areas contributes to the originality of this research. 

Author Response

Please see the file attached.

Reviewer 4 Report

Comments and Suggestions for Authors

This research demonstrates a method to detect the boundary of the metropolis. The paper gives a detailed description of the method. It is a minimum spanning tree like method, which is always used in model building in region research. However, there are some questions in this research, and it should have minor revisions.

The main contribution of this paper is that the paper could prevent ‘holes’ in optimization results. But there is no description of this word, and no cases could be seen. It is impossible for the reader to understand the background of the research.

The metropolis’s research should be cited. The background in part 2 is not enough to link the theory with the quantitative research.

For forces influencing the region’s detection, research should have more citations to explain the factors that could influence metropolis detection. 

For the applications, research should clearly divide the holes and the characteristics of Figure 8. For Bowang district, it is nearly a hole; the discussion of this case should analyze the difference between this part when it is considered as the hole and as a continuous part of another area.

Author Response

Please see the file attached.

Round 2

Reviewer 1 Report

Comments and Suggestions for Authors

The authors have improved the manuscript significantly. I recommend acceptance after minor revision.

1. Methods. Thank you for adding a reference to Shirabe’s flow model [43], which makes it easier to understand the proposed methods for ensuring continuity in spatial optimization. The advancement of this work over Shirabe’s seems to focus on ensuring the continuity of the internal and external regions simultaneously to ensure there is no hole in the internal (selected) area. It would be helpful to highlight this novelty in writing.

2. Figure 10. This is an interesting result. It may be more interesting if the authors could examine whether the total connection strength (blue line) plateaus with an increased number of counties. This may be beyond the scope of this paper.   

3. References for multi-core urban structures and urban percolation algorithms for your reference.

Cai, J., Huang, B., & Song, Y. (2017). Using multi-source geospatial big data to identify the structure of polycentric cities. Remote Sensing of Environment, 202(2016), 210–221. https://doi.org/10.1016/j.rse.2017.06.039

Arcaute, E., Molinero, C., Hatna, E., Murcio, R., Vargas-Ruiz, C., Masucci, A. P., & Batty, M. (2016). Cities and regions in Britain through hierarchical percolation. Royal Society Open Science, 3(4). https://doi.org/10.1098/rsos.150691

Comments on the Quality of English Language

4. Ln 136-138. Incomplete sentence.

Author Response

Please see the file attached.

Reviewer 2 Report

Comments and Suggestions for Authors

The authors answered my questions. Thanks!

Author Response

We appreciate your constructive comments and feedbacks.